# Biocompatible, Resilient, and Tough Nanocellulose Tunable Hydrogels

**DOI:** 10.3390/nano13050853

**Published:** 2023-02-24

**Authors:** Amir Rudich, Sunaina Sapru, Oded Shoseyov

**Affiliations:** Robert H. Smith Faculty of Agriculture, Food and Environment, The Center for Nano Science and Nano Technology, The Hebrew University of Jerusalem, Rehovot 76100, Israel

**Keywords:** cellulose nanocrystals, cellulose nanofibrils, resilient hydrogels, biocompatibility, polyacrylamide, grafting

## Abstract

Hydrogels have been proposed as potential candidates for many different applications. However, many hydrogels exhibit poor mechanical properties, which limit their applications. Recently, various cellulose-derived nanomaterials have emerged as attractive candidates for nanocomposite-reinforcing agents due to their biocompatibility, abundance, and ease of chemical modification. Due to abundant hydroxyl groups throughout the cellulose chain, the grafting of acryl monomers onto the cellulose backbone by employing oxidizers such as cerium(IV) ammonium nitrate ([NH_4_]_2_[Ce(NO_3_)_6_], CAN) has proven a versatile and effective method. Moreover, acrylic monomers such as acrylamide (AM) may also polymerize by radical methods. In this work, cerium-initiated graft polymerization was applied to cellulose-derived nanomaterials, namely cellulose nanocrystals (CNC) and cellulose nanofibrils (CNF), in a polyacrylamide (PAAM) matrix to fabricate hydrogels that display high resilience (~92%), high tensile strength (~0.5 MPa), and toughness (~1.9 MJ/m^3^). We propose that by introducing mixtures of differing ratios of CNC and CNF, the composite’s physical behavior can be fine-tuned across a wide range of mechanical and rheological properties. Moreover, the samples proved to be biocompatible when seeded with green fluorescent protein (GFP)-transfected mouse fibroblasts (3T3s), showing a significant increase in cell viability and proliferation compared to samples comprised of acrylamide alone.

## 1. Introduction

Hydrogels are three-dimensional hydrophilic crosslinked polymer networks that are highly absorbent while keeping their original structure [1,2,3,4]. These materials have been proposed as potential candidates for many different applications and fields, including artificial muscles, vertebral disk replacements, liquid and heavy metal absorbents, antibacterial materials, and as electroconductive materials [5,6,7,8,9]. However, many hydrogels exhibit poor mechanical properties: low extensibility, brittleness, low resilience, irreversible deformation, and lack of biocompatibility, which limit their applications [10,11,12].

Many approaches have been attempted to improve the mechanical properties of hydrogels, such as utilizing topological polymers, nanocomposites, microgels, and double network-based reinforcements [13,14,15,16,17]. Among these approaches, hydrogels with nanocomposite reinforcements have generated much attention [9,18,19,20]. Nanomaterials such as carbon nanotubes (CNTs), silica, clay, and other inorganic particles have been employed, producing some encouraging results [20,21,22,23,24], though mostly at the cost of biocompatibility loss and potential cytotoxicity [25,26,27].

Recently, the application of various cellulose-derived nano particles have emerged as attractive candidates as nanocomposite-reinforcing agents due to their biocompatibility, abundance, and relative ease of chemical modification [28,29,30,31]. Cellulose is the most abundant biopolymer on earth, and functions as the load-bearing component in plants’ cell walls. Cellulose may be degraded using mechanical shearing methods to cellulose nanofibrils (CNF), which are comprised of both the crystalline and amorphous regions of the cellulose fiber, resulting in particles which are 10–40 nm in width and span several microns in length. Further degradation, usually by acid hydrolysis, leads to the extraction of the crystalline regions alone, known as cellulose nanocrystals (CNC). CNCs are 5–10 nm in width and a few hundred nanometers in length, depending on the cellulose source.

Among the possible ways to integrate cellulose-derived nanomaterials as reinforcing components in hydrogels, polymer grafting onto the cellulose backbone has proven a versatile and effective option [32,33]. Due to the abundance of hydroxyl groups throughout the cellulose chain, cellulose-derived nanomaterials are ideal candidates for these types of chemical modifications.

The modification of cellulose particles by polymer grafting has been studied extensively and has been used to grant many different properties onto cellulose particles, such as increased dispersibility, fabrication of bio-degradable materials, agricultural nutrient release agent, antimicrobial material, highly resilient polymers, and increased processability [34,35,36,37,38,39]. Among the various techniques investigated, the formation of free radicals on the cellulose-based particles’ backbone surface through direct oxidation, using oxidizers such as Fenton reagents (Fe^2+^/H_2_O_2_) or ceric ions (Ce^4+^), has proven the most common and effective [30,33,40,41,42]. In this method, water-soluble oxidizers form free radicals on the cellulose backbone in the presence of acid in an aqueous medium, without requiring additional pretreatments [32]. When employing oxidizing salts such as cerium ammonium nitrate ([NH_4_]_2_[Ce(NO_3_)_6_], CAN), the radicals formed can initiate the graft polymerization of vinyl and acryl monomers, as illustrated in Figure 1. In addition, many acrylic monomers such as acrylamide (AM) may also readily polymerize by radical methods, yielding a graft polymerization system of covalently bound cellulose–acrylamide complexes, from which long polyacrylamide (PAAM) chains begin propagating from the particle’s surface [43].

Cerium-initiated graft copolymerization on cellulose-derived nanomaterials has been described elsewhere, for the fabrication of composite hydrogels for increased thermal and salinity tolerance, heavy metal absorbents, and for the improvement in mechanical properties such as stretchability, toughness, tensile strength, and resilience [42,45,46,47]. In this work, cerium-initiated graft polymerization was applied to cellulose-derived nanomaterials, namely CNC and CNF, in a polyacrylamide matrix to fabricate nanocomposite hydrogels which display high resilience, tensile strength, and extensibility. This work proposes that by introducing mixtures of different ratios of CNC and CNF, a whole range of rheological and mechanical properties can be achieved. These properties may prove valuable for different fabrication technologies such as molding and additive manufacturing. Copolymerization of the composite is achieved by the propagation of acrylamide chains grafted from the cellulose-derived nanoparticle’s surface by radical polymerization and heat, forming polyacrylamide (PAAM). Once propagated, the long chains are further crosslinked by adding N,N’-methylene bisacrylamide (BAM) to form a uniform chemical network of long flexible chains interconnected by homogeneously dispersed cellulose-derived nanomaterials, which function both as a crosslinker and as a reinforcing agent, resulting in a highly extendable, resilient soft polymer (Figure 2), with biocompatible properties, crucial for any envisioned biomedical application.

## 2. Materials and Methods

### 2.1. Materials

Cellulose nanocrystals were supplied by Melodea Ltd. (Rehovot, Israel); cellulose nanofibrils were supplied by Cellulose Lab (Fredericton, New Brunswick, Canada). Acrylamide (AM), N,N’-methylene bisacrylamide (BAM), and cerium(IV) ammonium nitrate ([NH_4_]_2_[Ce(NO_3_)_6_],CAN) were purchased from Sigma-Aldrich. Hydrochloric acid (HCl) was purchased from Romical (Beer Sheva, Israel).

### 2.2. Hydrogel Fabrication

Hydrogels were fabricated based on cerium-initiated graft copolymerization by employing ceric ions to initiate polymerization of acrylamide grafted from the cellulose particle backbone. CNC:CNF mixture suspensions were prepared by mixing CNC and CNF suspensions (3 wt% each) at 1:3, 1:1, and 3:1 (wt/wt) CNC to CNF and homogenized by sonication using a Q500 ultrasonic processor (Qsonica, Newtown, CT, USA) at 80% amplitude 1 s on/1 s off pulse regime for a total of 30 min. The CNC:CNF solutions were then diluted to 1.6 wt% with AM (0.07 mol) and BAM (0.03 mmol) to a total volume of 20 mL, as is suggested by the protocol described by Weng et al. [47]. Next, an initiator solution was prepared by dissolving CAN (0.036 mmol) in 5 mL double distilled water and adjusting the pH to 1 using 1 M HCl. The initiator solution was added dropwise to the cellulose-acrylamide mixture by stirring and poured into molds. Polymerization occurred at 50 °C for 12 h. Fabricated hydrogels were carefully removed for further testing. The resulting series of compositions were labeled according to their CNC to CNF content ratio, and categorized, namely C1F3 (1:3), C1F1 (1:1), C3F1(3:1), C1F0 (1:0), and C0F1 (0:1), where C and F represent CNC and CNF, respectively.

### 2.3. Mechanical Testing (Tensile)

Tensile tests were performed using a tensiometer (Instron 3345 tester, Norwood, MA, USA) equipped with a 100 N load cell, using a crosshead speed of 10 mm/min on dumbbell samples (length = 65 mm; width = 10 mm; thickness = 2 mm; gauge length = 10 mm; inner width = 4 mm). Dimension measurements were conducted by a digital caliper (0.1 mm resolution). Each sample was elongated until failure. Samples were tested as prepared, and surface coated with silicone oil to minimize water evaporation during testing. Stress–strain curves were plotted based on 4 repeats of each composition. Sample elongation and ultimate tensile stress (UTS) were recorded, toughness was defined as the area under the extension curve, and the Young’s modulus was calculated by the slope of the linear elastic region.

Cyclic tensile tests were performed using a tensiometer (Instron 3345 tester, Norwood, MA, USA) equipped with a 100 N load cell, using a crosshead speed of 100 mm/min on dumbbell samples (length = 65 mm; width = 10 mm; thickness = 2 mm; gauge length = 10 mm; inner width = 4 mm). Dimension measurements were conducted by a digital caliper (0.1 mm resolution). Samples were tested as prepared, and surface coated with silicone oil to minimize water evaporation during testing. Each sample was subjected to a 4-cycle extension at 100% increments until reaching mechanical failure. Resilience was calculated from the second, third, and fourth extensions of each cycle, by the ratio between the areas under the unloading and loading curves.

### 2.4. Rheology

The rheological properties of fabricated hydrogels were evaluated using a Kinexus PRO+ rheometer (Netzsch, Selb, Germany). Tests were conducted using parallel plates 40 mm in diameter at 25 °C. Solutions composed of different CNC:CNF ratios before polymerization were evaluated by viscosity as a function of shear rate measurements, performed by ramping the shear rate from γ̇ =0. 1 1/s to γ̇ = 1000 1/s at 25 °C, using 40 mm plates with a plate gap of 0.5 mm.

Polymerized hydrogel samples were kept in phosphate-buffered saline (PBS, 150 mM NaCl, 2.5 mM KCl, 10 mM K_2_HPO_4_, 2 mM KH_2_PO_4_, pH7.4) until fully hydrated, reaching constant weight, followed by frequency and amplitude sweeps. Frequency sweeps were preformed from 0.1 to 100 Hz at a fixed strain of 1%, and the storage (G′) and loss (G″) moduli were recorded. Amplitude sweeps were performed at a constant 1.33 Hz, at a sheer strain (γ̇(%)) in ranges between 0.1 and 100%. The linear viscoelastic region (LVER), yield point (τ_y_), where the applied stress begins to cause damage to the structure, flow point (τ_f_), and the flow transition index (τ_f_/τ_y_) have been calculated in all samples.

### 2.5. Swelling and Degradation Studies

Swelling and degradation tests were conducted in physiological conditions by submerging hydrogel specimens in phosphate-buffered saline (PBS, 150 mM NaCl, 2.5 mM KCl, 10 mM K_2_HPO_4_, 2 mM KH_2_PO_4_, pH 7.4). For swelling analysis, samples were incubated in PBS for a fixed duration and weighed after gently removing the excess liquid from the surface. The samples were returned to PBS until the next reading. The procedure was repeated until a plateau was achieved for the swelling.

Degradation of the samples was determined by incubating the samples in PBS for 6 weeks, regularly changing PBS and calculating the dry weight of the samples once per week. Prior to weighing, each sample was washed using DI water to remove excess salts from the material. Both experiments were performed in triplicate.

### 2.6. Cell Culture

Maintenance of animal cells and sample preparation for biocompatibility studies:

Green fluorescent protein (GFP)-transfected mouse fibroblasts (3T3s) were generously donated by Dr. Sharon Elizur-Schlesinger (The Hebrew University of Jerusalem). Cells were cultured in Dulbecco’s Modified Eagle’s Medium (DMEM) containing L-glutamine, glucose, sodium pyruvate, and sodium bicarbonate. Media were supplemented with fetal bovine serum to a final concentration of 10% and antibiotics, i.e., penicillin (10 Units/mL) and streptomycin (10 µg/mL), to minimize possible contamination. The cells were maintained in a humidified environment of a sterile cell culture incubator maintained with 5% carbon dioxide at 37 °C. The media of the cultured cells were regularly changed, and cells were passaged after achieving 70% confluency.

The hydrogels were sterilized with 70% ethanol before cell seeding. In brief, the fabricated hydrogels were incubated in 70% ethanol for 30 min and irradiated with UV simultaneously in a sterile cell culture hood. Samples were subsequently washed thrice with sterile phosphate-buffered saline (PBS, calcium, and magnesium free, pH 7.4) with a 20 min incubation in each wash to remove any traces of ethanol. Then, samples were acclimatized with DMEM media overnight. The media was removed, and hydrogels were dried for 4 h in a sterile cell culture incubator before adding the cells to enhance the efficiency of static cell seeding (5 × 10^4^ cells/hydrogel) onto the porous hydrogel.

### 2.7. Analysis of Cellular Toxicity and Proliferation

Proliferation and morphology of the cells onto the fabricated hydrogels was evaluated using a confocal microscope. As the 3T3s were transfected with GFP, no additional staining was conducted. A week post cell seeding, the cell-laden hydrogels were fixed with 4% paraformaldehyde, washed thrice with PBS for 10 min each, and observed under a confocal microscope (Leica Stellaris 5 FLIM STED with a climate incubator, Leica, Wetzlar, Germany). The area covered (%) by the cells with respect to the total area in each hydrogel composition was calculated by ImageJ using the confocal images obtained post culturing the cells for a week. Cellular migration was measured for composition C1F3 through the 3D view observed in confocal microscopy. Cellular proliferation and viability were determined for the cell-laden hydrogels by a 3-(4,5-dimethylthiazol-2-yl)-2,5-diphenyltetrazolium bromide (MTT) assay that calculates the metabolic activity of the cells and provides an indicator for cellular viability. The procedure followed for the assay was as per the manufacturer’s protocol. In brief, the hydrogels were transferred to fresh wells to avoid any reading from cells that were not attached to the hydrogels. These cell-laden hydrogels were then incubated in MTT dye for 4 h. The purple-colored product formed upon the reduction in MTT dye by cellular oxidoreductase is dissolved in dimethyl sulfoxide (DMSO), and the colorimetric reading was obtained in a spectrophotometer at 595 nm.

## 3. Results

### 3.1. Hydrogel Fabrication

Composite hydrogels were fabricated by redox-initiated free radical graft polymerization of AM and cellulose-derived particles. First, grafting of the AM onto the particles’ surface was initiated using Ce4+ as an oxidizer. This generates free radicals that form on the surface of the cellulose-derived particle, leading simultaneously to the formation of grafted cellulose particles and to the propagation of polyacrylamide (PAAM) chains from dissolved AM monomers. Propagating chains are then further crosslinked with BAM, which leads to the formation of the uniform hydrogels. The resulting hydrogels ranged from transparent when integrating CNC alone (C1F0) to opaque when comprising CNF only (C0F1) (Figure 3).

### 3.2. Tensile Testing

Mechanical properties of the fabricated hydrogels were evaluated by tensile tests. Typical tensile stress–strain curves are presented in (Figure 4), and measured mechanical properties are summarized in (Table 1). Composites fabricated with the integration of CNC alone (C1F0) expressed moderate UTS, recorded at 107 kPa, and an elongation at break of 441%, while displaying a Young’s modulus of 23 kPa. Conversely, composites fabricated with the integration of CNF alone (C0F1) resulted in a hard and non-uniform gel, displaying a three-fold increase in Young’s modulus, recorded at 307 kPa, and a high UTS of 390 kPa, but extended by only 105%, the lowest value recorded. It shows that the ratio of fibrils to crystals has a high influence on aspect ratio, rheological properties, and colloidal stability. The incorporation of a 1:3 ratio of CNC to CNF (C1F3) expressed an increase in elongation, UTS, and toughness as compared to C0F1, recorded at 229%, 527 kPa, and 624 kJ/m^3^, respectively. C3F1 resulted in a highly extendable gel, capable of reaching 687% extension and achieving a UTS of 483 kPa. This composition yielded the toughest hydrogel of all samples tested, reaching a calculated toughness of 1927 kJ/m^3^. These results suggest that by tailoring the ratios of added CNC or CNF, a range of mechanical performance is displayed, indicating the tunable nature of the mechanical properties of the fabricated hydrogels. A possible explanation can be attributed to the physical properties of both constituents; CNC displays strong colloidal stability and shear-thinning properties, which increase the homogeneity of the dispersion aiding in the formation of a uniform network. CNF possesses higher aspect ratio (i.e., particle length), which grants increased mechanical performance, but tends to clump together and aggregate, leading to non-uniform densities throughout the network and increased viscosity, which hinder processing possibilities, casting, and shaping of the material.

### 3.3. Cyclic Tensile Testing

Resilience and extension profiles were evaluated by cyclic tensile tests for four consecutive cycles. All samples evaluated followed a two-regime extension behavior. The first cycle shows a plastic deformation region, and the following three cycles display near perfect elasticity. As summarized in (Table 2), the resilience of each sample was calculated at two points for all samples; first, following 200% extension, which was the lowest common denominator reached by all samples tested. Second, the resilience was evaluated at the maximum extension cycle reached by each individual sample. At 200% extension (Table 2, column A), all samples showed impressive resilient characteristics, reaching close to or upwards of 90%, with low error between repeating cycles. Samples with high CNC content, C1F0, C1F1, and C3F1, expressed exceptionally high resilience, recorded at 91.5%, 92.5%, and 93.6% respectively. Samples with high CNF content, C0F1, and C1F3 expressed slightly reduced resilient properties, recorded at 87.3% and 89%, respectively. When comparing resilience at maximum extension reached by each individual sample (Table 2, column B), all samples displayed a slight reduction in resilience compared to the values calculated at 200%, possibly due to an accumulation of plastic deformation which led to increased energy loss. A few key differences between the ratios tested can be seen. First, C1F0 (Figure 5d) began expressing high resilience, 91.5% at 200% extension, but experienced a 10.3% decrease in resilience upon reaching its maximum recorded extension of 1000%, down to 82.2%, pointing to material fatigue over the course of repeating extension cycles at increasing strains. C0F1 (Figure 5e) exhibited the lowest resilience recorded at 200% extension, reaching 87.3%. This was also the maximum extension reached by this composition, thus resulting in a material expressing relatively low extensibility and resilience. When comparing samples composited with both CNC and CNF, all samples showed exceptional resilience at 200% extension, with C1F3, C1F1, and C3F1 (Figure 5a–c) exhibiting 89%, 92.5%, and 93.6% resilience, respectively. All these compositions expressed about a 2% reduction in resilience between the 200% extension cycle and their respective maximum extension, reaching 87%, 90%, and 90.5% respectively. To emphasize the elastic nature of the material, C3F1 exhibited 90.5% resilience at an 800% extension, following 32 extension cycles at increasing strains. This points to the resulting composites possessing a highly resilient nature at a wide range of mechanical strains and that the maximum extendibility and ultimate stress can be fine-tuned by incorporating different compositions of CNC and CNF. This may arise from CNC enabling better dispersion throughout the matrix, resulting in a more uniform and better-defined network upon polymerization as compared to CNF. It is also possible that the use of CNF leads to an increased network density.

### 3.4. Rheological Properties

#### 3.4.1. Viscosity Measurements

Viscosity as a function of shear rate was evaluated to quantify the viscous behavior of the different solutions when incorporating varying ratios of CNC and CNF. Cellulose-derived nanomaterials generally, including CNC, possess known shear-thinning properties [28,48,49]. Hence, optimizing the CNC:CNF ratio was expected to yield a reduction in overall viscosity compared to pure CNF. Reduced viscosity may aid in the homogenization, handling, and processing of the pre-polymerized solutions and the homogeneity of resulting samples, leading to consistent mechanical performance and sample reproducibility. As the resulting flow curves suggest (Figure 6), a reduction in viscosity occurred in all samples as shear rates increased. However, C0F1 had the highest measured viscosity and plateaus between the shear rates of γ̇ = 10^1^ (s/1) and γ̇ = 10^2^ (s/1), at which point the CNF particles aggregated and the solution appeared to separate into denser and thinner portions. On the other hand, C1F0 expressed lower initial viscosity and expressed shear-thinning behavior expected from CNC-based suspensions. CNC and CNF mixed samples, such as C1F3, with relatively low CNC content, had lower viscosity than C0F1, while no phase separation was observed throughout the measurement. Samples C1F1 and C3F1 expressed similar viscosities to C1F0, showing the ability to incorporate CNF without compromising the solution’s viscosity, allowing effective mixture and dispersion, and resulting in homogeneous samples upon polymerization.

#### 3.4.2. Frequency Sweeps

Frequency sweeps were conducted to evaluate the structural integrity of the fabricated hydrogels (Figure 7). Samples demonstrated a largely frequency-independent modulus with G′ > G″, where the storage (G′) and loss (G″) moduli followed a similar trend, comparable with permanently crosslinked gels. Furthermore, there was an increase in storage modulus with increasing ratios of CNF. Comparing samples composed of both CNC and CNF to those incorporating CNC or CNF alone leads to the following conclusions. C1F0 samples resulted in the softest composite, expressing the lowest storage modulus recorded (G′ = 3 kPa). On the other hand, C0F1 expressed a relatively high storage modulus (G′ = 24 kPa) (Figure 7a), although only half the storage modulus was measured for C1F3 (G′ = 50 kPa) (Figure 7b). This may arise from the inhomogeneity of the composites based on CNF alone. These have inconsistent densities, leading to less stable structures overall. CNC eases the formation of a homogeneous dispersion and increases uniformity throughout the fabricated hydrogel, which benefits the structural integrity of the sample. CNF enables the formation of tougher, more rigid structures compared to CNC-based composites, indicated by increased storage moduli achieved in C1F3 samples. Even by the addition of relatively small amounts of CNF in sample C3F1, a higher storage modulus was reached compared to samples incorporating CNC alone (C1F0), recorded at 10 kPa as opposed to 3 kPa, respectively.

#### 3.4.3. Amplitude Sweeps

Amplitude sweeps were employed to describe the rheological stability of the composites. The linear viscoelastic region (LVER) of samples incorporating compositions of CNC and CNF (C1F3, C1F1, and C3F1) was increased compared to samples comprising CNC or CNF alone (Table 3), indicating increased structural stability overall. Next, the flow points (τ_f_) of all samples followed similar trends to the LVERs calculated. Here, C1F3 (Figure 8a) and C3F1 (Figure 8c) outperformed others, reaching the flow transition point of γ̇ = 31.6%, indicating an ability to withstand a wider range of deformations compared to other compositions. Finally, the flow transition index, which is derived from the ratio between the flow point and yield point (τ_y_), refers to the area in which the LVER has passed but the amount of deformation has yet to arrive at a critical point where the micro-structure begins to crack, where G″ > G′. It can be determined that C0F1 (Figure 8e) showed the poorest structural integrity, expressing the lowest flow transition index measured at 10. C1F0 (Figure 8d), on the other hand, showed a relatively high flow transition index of 59.9. This may show that by employing CNC, the material is able to withstand a higher amount of deformation, possibly due to the small size of the CNC particles, which act as efficient, small crosslinking joints and stress recovery mechanisms. C1F3 (Figure 8a) and C1F1 (Figure 8b) expressed a higher flow transition index compared to C0F1 but lower than C1F0, recorded at 39.8 and 31.6, respectively. This may point to CNF still governing much of the mechanical structure of the samples in these compositions. The role of CNC, which we theorize aids as a rheological modifier, appears to be more pronounced in C3F1 (Figure 8c), which exhibits the highest flow transition index of all samples evaluated, recorded at 63.06.

### 3.5. Swelling and Degradation Profiles

#### Swelling Analysis

The swelling properties of hydrogels may affect their mechanical performance, and their assessment can aid in determining the material’s potential applications and roles. As many potential functions for hydrogels are intended for biomechanical applications, PBS at physiological pH was chosen as a swelling buffer. All compositions tested retained their original shape and geometry throughout the testing period (>6 weeks) when submerged in excess PBS without dissolving or rupturing, implying a strong crosslinked network. C1F0 samples swelled the most, swelling to 1150% of their original weight over a 5-day period. C0F1 samples swelled the least, reaching 750% of their original weight over a 5-day period, possibly pointing to increased network density resulting from the longer, fibrillated CNF particles. The swelling properties of CNC:CNF blends generally related to the ratio of their constituents: an increased CNC ratio led to an increased swelling ratio, as can be seen in sample C3F1, while an increased CNF ratio led to a decrease in swelling capabilities (Figure 9a). Samples which incorporated higher loadings of CNF reached their swelling equilibrium faster, as can be seen in samples C0F1 and C1F3, equilibrating after 10–15 h as opposed to samples with higher CNC content, C1F0 and C3F1, which reached equilibration after 60 h.

As stated above, many possible applications for hydrogels are intended to function in aqueous environments or, more importantly, in biologically active environments such as the human body, and may become exposed to several degrading factors. Thus, it is important to determine the possible degradation of the material. Degradation may lead to residual material circulating in the body, potentially causing undesired effects such as clotting, creating blockage, and inflammation. Results suggest that most tested samples kept most of their original weight (>95%) throughout the 6-week period evaluated. Composites fabricated by the incorporation of CNF alone, C0F1, lost some weight (~10%) following the first week tested, which may result from sample inhomogeneity, leading to un-crosslinked residues leeching out of the sample (Figure 9b).

### 3.6. Biocompatibility and Cytotoxicity Studies

Preliminary screening of the fabricated hydrogels for their potential as a biomaterial was conducted by evaluating their cytocompatibility and toxicity with mammalian fibroblasts. The GFP-transfected 3T3s (GFP-3T3s) were statically seeded on the hydrogels, while media were added to the wells containing the hydrogels after an hour to allow time for maximum cellular adhesion. It is vital for any biomaterial to be nontoxic and biocompatible. A preliminary evaluation of CNC, CNF, CNC:CNF blends, and a control group comprised of acrylamide (AM) alone without the addition of cellulose-derived nanomaterials was conducted. Hydrogels seeded with the GFP-3T3s cells indicated that all samples fabricated with the addition of the cellulose-derived nanomaterials supported the adhesion and proliferation of mammalian cells, as can be seen in the confocal microscope images (Figure 10a). MTT analysis indicated a significant difference in cellular adhesion in all hydrogels when compared with the control AM hydrogel. No significant difference was observed amongst the test samples (C1F1, C3F1, and C1F3) for cellular adhesion as observed on day 1 post cell seeding, indicating that all hydrogels fabricated based on mixtures of CNC and CNF exhibited similar properties suitable for cell adhesion (Figure 10b). However, 7 days post cell seeding, sample C1F3 showed significant growth in cells compared to C1F1 and C3F1 (Figure 10b). CNF appeared to contribute to cellular proliferation compared to the addition of CNC, which is indicated by a significant difference in cellular growth when comparing samples C0F1 and C1F0 following incubation for 7 days (Figure 10b). The area covered by the fibroblasts calculated from the confocal images also reveals that C1F3 had significantly higher cell coverage compared to samples C1F1 and C3F1 (Figure 10c). Finally, morphological observations revealed that sample C1F3 exhibited a more uniform cellular growth with widespread spindle-shaped morphology (Figure 10a) and relatively deep cell migration (~80 µm) (Figure 10d). These preliminary evaluations indicate that all hydrogels fabricated based on the incorporation of mixtures of CNC:CNF (C1F1, C1F3, and C3F1) showed improved cellular behavior compared to samples fabricated with the incorporation of only one of the cellulose-derived nanomaterials (C1F0 and C0F1) or without the addition of any cellulose nanomaterials (AM). This makes it promising as a biomaterial, with C1F3 displaying the most suitable properties for cellular adhesion and proliferation compared to the other composites, based on CNC:CNF mixtures evaluated.

## 4. Conclusions

This study has shown the fabrication of hydrogels based on the incorporation of two cellulose-derived nanomaterials in a cerium-initiated graft polymerization system with acrylamide. By combining the two cellulose-derived particles, CNC and CNF, the material fabricated has proven to be highly tunable across a wide range of mechanical properties and rheological performance. We were able to increase the material’s UTS 5-fold, from ~0.1 MPa to ~0.5 MPa, and its toughness 10-fold, from ~0.2 MJ/m^3^ to ~1.9 MJ/m^3^, between the integration of CNC alone and a mixture of both CNC and CNF, while maintaining a highly resilient structure (~92%). CNC’s dispersibility and shear-thinning properties lead to better homogeneity of the pre-polymerized solution, as is evident from viscosity measurements and throughout the polymerized samples, resulting in samples expressing increased stretchability and resilience. The incorporation of CNF, on the other hand, resulted in tougher samples, as is evident in the tensile tests. This comes at the cost of poor sample homogeneity. The combination of both cellulose-derived particles leads to a decrease in overall solution viscosity compared to solutions with CNF alone, which aids in the homogeneous dispersion of the nanofibrils throughout the matrix before polymerization. Comparing similar reported systems based on the grafting of acrylic polymers onto cellulose-derived nanomaterials points to an improvement in tensile and resilient properties [38,42,50,51,52]. Finally, the materials’ potential as a biomaterial, supplying a suitable environment for the viability and proliferation of fibroblasts, without the requirement of integrating cell attachment-promoting elements such as the arginyl glycosylaspartic acid (RGD) peptide, has been shown. The combination of impressive tunable mechanical performance and biocompatible properties may point to the fabricated composite as a promising candidate for biomedical and scaffolding applications.

## Figures and Tables

**Figure 1 nanomaterials-13-00853-f001:**
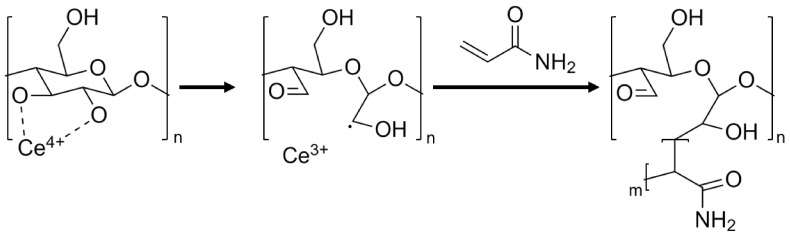
Proposed mechanism of cerium-initiated copolymerization of cellulose and acrylamide [43,44].

**Figure 2 nanomaterials-13-00853-f002:**
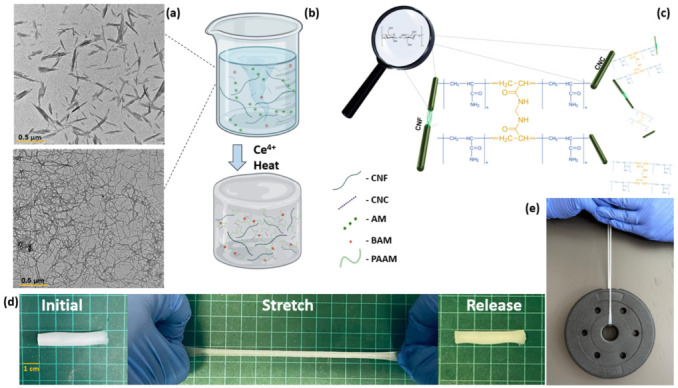
Schematic hydrogel fabrication overview and properties. (**a**) Transmission electron microscope (TEM) micrographs of CNC and CNF (reproduced with permission form copyright owner). (**b**) Experimental overview of hydrogel fabrication. (**c**) Proposed microstructure of polyacrylamide complexes grafted onto CNC and CNF particles crosslinked by N,N’-methylene bisacrylamide. (**d**) Photographs depicting the stretchability of fabricated hydrogels. (**e**) Photograph of a 1.5 kg weight held by the hydrogel sample.

**Figure 3 nanomaterials-13-00853-f003:**
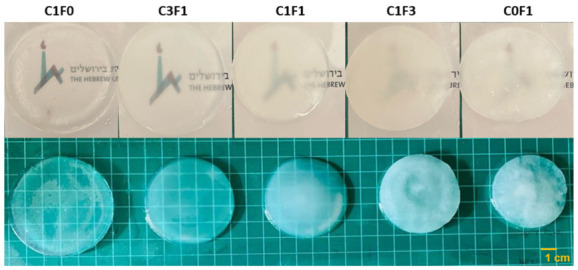
Illustrated overview of the fabricated hydrogels and their opacity in variable compositions. C1F0 fabricated with the addition of CNC alone, C0F1 fabricated with the addition of CNF alone, and C3F1, C1F1, and C1F3 fabricated with CNC and CNF at 3:1, 1:1, and 1:3, respectively.

**Figure 4 nanomaterials-13-00853-f004:**
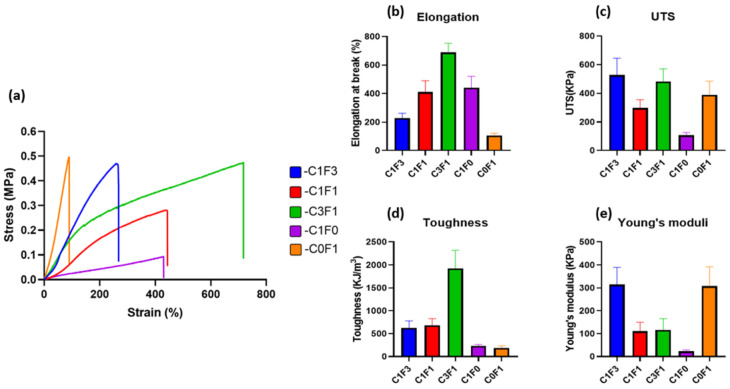
Mechanical properties (tensile) of the fabricated cellulose-based hydrogels. (**a**) Tensile stress–strain curves, (**b**) elongation at break, (**c**) ultimate tensile stress, (**d**) toughness, and (**e**) Young’s modulus of different hydrogels fabricated with variable composition of CNC and CNF.

**Figure 5 nanomaterials-13-00853-f005:**
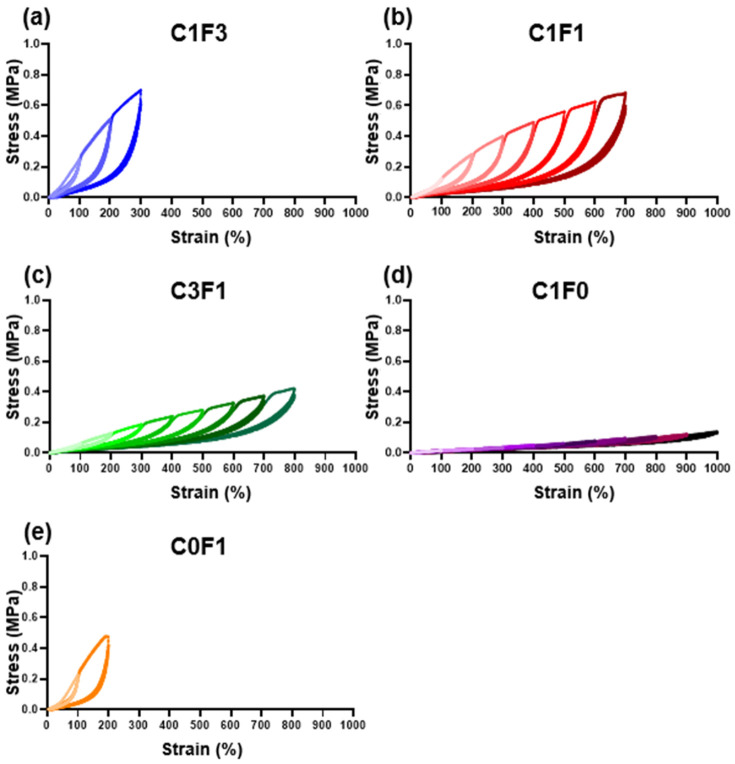
Cyclic tensile loading. Cyclic tensile tests of different fabricated hydrogel samples, namely (**a**) C1F3, (**b**) C1F1, (**c**) C3F1, (**d**) C1F0, and (**e**) C0F1, with CNC and CNF in variable compositions.

**Figure 6 nanomaterials-13-00853-f006:**
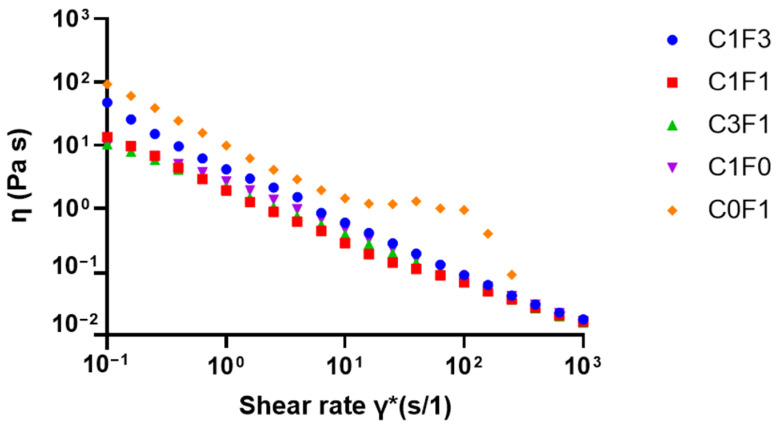
Viscosity as a function of shear rate for solutions with varying ratios between CNC and CNF before polymerization.

**Figure 7 nanomaterials-13-00853-f007:**
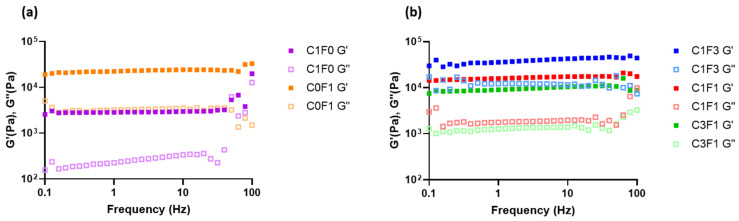
The rheological measurements of the fabricated hydrogel composites represented in loss modulus G″ and storage modulus G′ with respect to frequency for samples C1F0 and C0F1 (**a**) and C1F3, C1F1, and C3F1 (**b**).

**Figure 8 nanomaterials-13-00853-f008:**
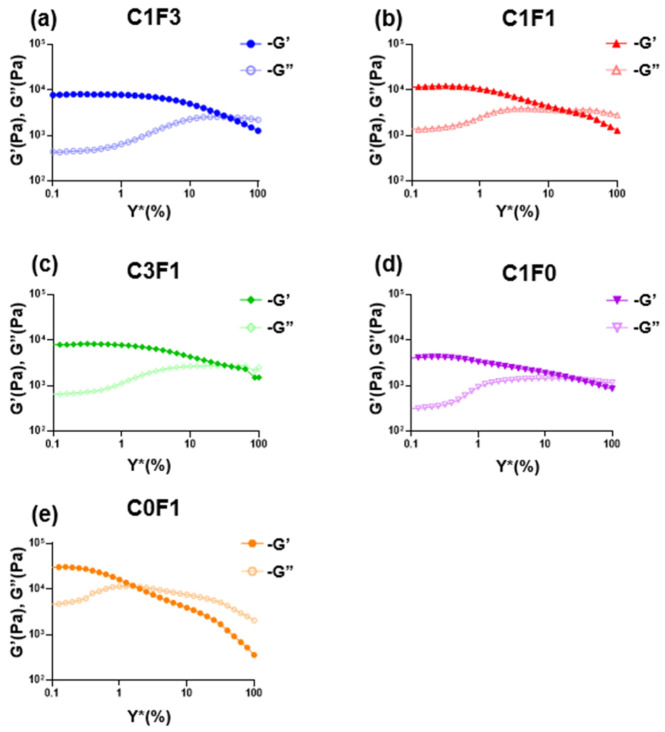
The rheological measurements of the fabricated hydrogel composites represented in loss modulus G″ and storage modulus G′ with respect to amplitude for different fabricated hydrogel samples, namely (**a**) C1F3, (**b**) C1F1, (**c**) C3F1, (**d**) C1F0, and (**e**) C0F1, with CNC and CNF in variable compositions.

**Figure 9 nanomaterials-13-00853-f009:**
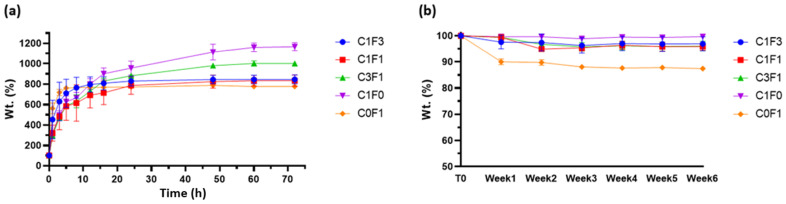
(**a**) Swelling profiles and (**b**) degradation profiles of fabricated hydrogels submerged in physiological PBS at pH 7.4.

**Figure 10 nanomaterials-13-00853-f010:**
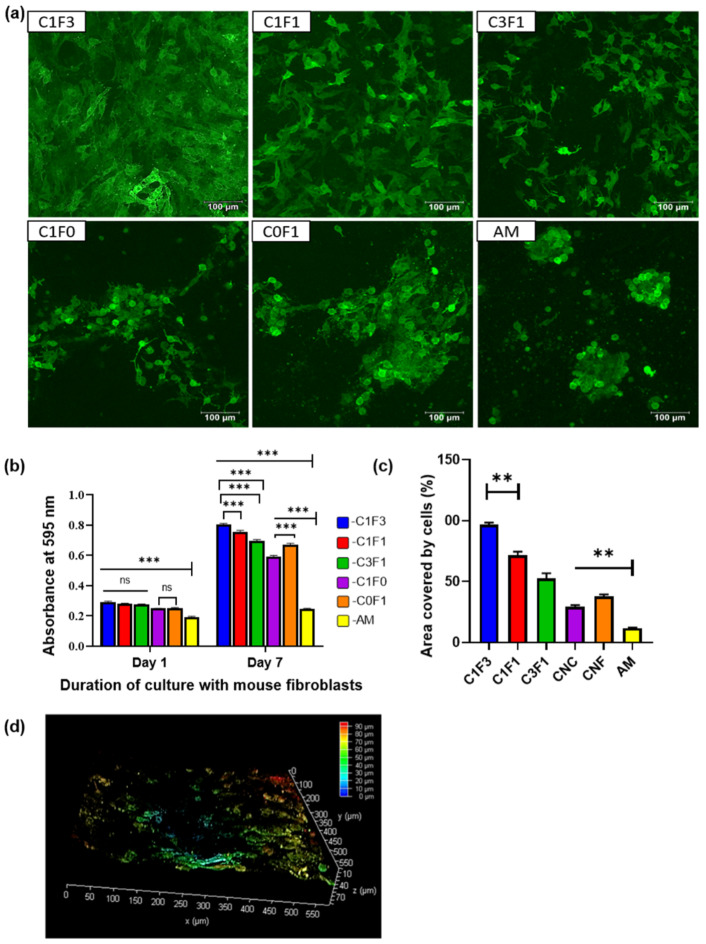
Cytocompatibility of CNC–CNF and acrylamide-based hydrogels with mouse fibroblasts, 3T3s (transfected with GFP). (**a**) Confocal micrographs taken after culturing 3T3s-GFP onto the hydrogels for a week. The images are representative images of overlapped z-sections. (**b**) Confocal image for sample C1F3 represented in 3D view depicting the depth coverage by the cells. (**c**) Area covered by cells in individual hydrogels calculated from the confocal images using ImageJ. Statistical analysis was performed by the Tukey test, where ** represents *p* < 0.001 (n = 3). (**d**) Estimation of cellular proliferation and viability determined by the MTT assay with cell-laden hydrogels after day 1 and day 7 of culture with mouse fibroblasts. Statistical analysis performed by 2-way ANOVA, where *** and ns represent *p* < 0.001, and *p* = not significant, respectively (n = 3).

**Table 1 nanomaterials-13-00853-t001:** Calculated mechanical properties attained by tensile tests of fabricated hydrogels.

Sample Name	Elongation at Break (%)	Ultimate Tensile Stress (kPa)	Toughness (kJ/m^3^)	Young’s Modulus (kPa)
C1F3	229 ± 33	527 ± 117	624 ± 154	313 ± 75
C1F1	410 ± 79	298 ± 56	678 ± 147	110 ± 39
C3F1	687 ± 65	483 ± 87	1927 ± 390	116 ± 48
C1F0	441 ± 79	107 ± 19	229 ± 33	23 ± 6
C0F1	105 ± 17	390 ± 94	186 ± 47	307 ± 85

**Table 2 nanomaterials-13-00853-t002:** Calculated resilience values of fabricated hydrogels at 200% extension (A) and at maximum extension achieved by each composition individually (B).

Sample name	(A) Resilience(%) at 200% Extension	(B) Resilience(%) at Max Extension
C1F3	89 ± 1	87 ± 1
C1F1	92.5 ± 0.8	90 ± 0.7
C3F1	93.6 ± 0.7	90.5 ± 0.4
C1F0	91.5 ± 0.2	82.2 ± 0.1
C0F1	87.3 ± 1.4	87.3 ± 1.3

**Table 3 nanomaterials-13-00853-t003:** Calculated linear viscoelastic region (LVER), flow point, and flow transition index of fabricated hydrogel compositions.

Sample Name (CNC:CNF)	Linear Viscoelastic Region (LVER(%))	Flow Point (τ_f_(%))	Flow Transition Index (τ_f_/τ_y_)
C1F3	0.8	31.6	39.8
C1F1	0.5	15.8	31.6
C3F1	0.5	31.6	63.06
C1F0	0.2	11.9	59.9
C0F1	0.15	1.6	10

## Data Availability

Data is available on request from the corresponding author.

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
