# Peer review of "Biocompatible, Resilient, and Tough Nanocellulose Tunable Hydrogels"

_nanomaterials, 2023, doi:10.3390/nano13050853_

Round 1

Reviewer 1 Report

The research paper is related to the fabrication of biocompatible hydrogels from CNF and/or CNC and polyacrylamide, in order to enhance their mechanical, high resilience and extensibilityThe methodology, discussion and presentation of results is suitable, however, some minor changes are suggesting before it can be accepted for publications:

1.     The research was performed using cellulose nanofibrils or nanofibers? If, nanofibrils have been used, as predicted, the notation has to be changed throughout the whole manuscript. 

2.     Page 2 lines 65-71: CNCs are usually smaller than 1 micro-m, please check

3.     The “ nanoparticles” should be changed to “nanomaterials”

4.      The mechanical, cycle tensile testing and rheological study of polymerized samples have been performed only for dry samples. Since the samples swell in PBS, and are intend for biomedical application,  it would be useful that some measurements  would be performed also in a wet/swellable state. 

5.     SEM images, at least for a representative sample, before/after swelling would be welcome to see the homogeneity and micro/nanostructure

6.     The discussion could be improved by comparing the properties with similar hydrogels

Author Response

Dear reviewer

Thank you for taking the time to review our manuscript. We appreciate your constructive feedback and have revised our work according to your comments. Your insights are valuable and help us improve the quality of our research. In the following document, we will reply point-by-point (in red) to your comments (in italics).

The research paper is related to the fabrication of biocompatible hydrogels from CNF and/or CNC and polyacrylamide, in order to enhance their mechanical, high resilience and extensibilityThe methodology, discussion and presentation of results is suitable, however, some minor changes are suggesting before it can be accepted for publications:

  1. The research was performed using cellulose nanofibrils or nanofibers? If, nanofibrils have been used, as predicted, the notation has to be changed throughout the whole manuscript.
  • As suggested, the text has been adjusted accordingly to clarify the use of nano fibrils.
  1. 2.Page 2 lines 65-71: CNCs are usually smaller than 1 micro-m, please check
  • As suggested, the text has been adjusted accordingly (lines 78-80).
  1. 3.The “ nanoparticles” should be changed to “nanomaterials”
  • As suggested, the text has been adjusted accordingly.
  1. The mechanical, cycle tensile testing and rheological study of polymerized samples have been performed only for dry samples. Since the samples swell in PBS, and are intend for biomedical application,  it would be useful that some measurements  would be performed also in a wet/swellable state.
  • Tensile and cyclic tensile tests have been performed on “as prepared” samples (i.e., in a wet state though not at their maximum swelling capacity).
  • Rheological measurements on polymerized hydrogel samples (i.e., the frequency and amplitude sweeps) were performed on fully hydrated samples.
  • Both these points have been reiterated in the revised manuscript in the materials and methods section to better reflect these points.
  1. SEM images, at least for a representative sample, before/after swelling would be welcome to see the homogeneity and micro/nanostructure
  • We agree a thorough examination of the composite's microstructure to gain a more complete understanding of its properties would be welcome. However, due to current technical limitations, we will not be able to perform these tests in the current given timeframe.
  1. The discussion could be improved by comparing the properties with similar hydrogels
  • The discussion has been expanded upon to better reflect the results of the article while some note has been given to address and compare similar reported systems along with referenced recent literature.

Reviewer 2 Report

In this manuscript, authors fabricated biocompatible, resilient, and tough nanocellulose tunable hydrogels by introducing mixtures of differing ratios of CNC and CNF. The manuscript is also well-organized. However, there are still some issues to be addressed. A moderate revision is suggested before its acceptance.

1.     The background of this work should be shortened while more solid data should be added into the abstract section.

2.     More introduction on the structures, properties, and applications of nanocellulose and its related hydrogels should be further clarified with some recent supporting articles: Journal of Bioresources and Bioproducts, Volume 7, Issue 4, November 2022, Pages 245-269; Hydrochar-embedded carboxymethyl cellulose-g-poly(acrylic acid) hydrogel as stable soil water retention and nutrient release agent for plant growth; etc.

3.     One scheme to present the experimental procedure should be added for better understanding.

4.     Some of the figures should be modified with better readability, especially the texts.

5.     In Fig.5, the labels in (d) and (e) do not agree with what is written above, and the research object in (f) is not marked.

6.     Figures should be places under the contents when the figures were firstly mentioned.

7.     In table 3, the meaning of τy is not explained in the text.

8.     The spindle shaped morphology mentioned in the article is not clearly seen in Fig.10.

9.     Authors have introduced the definition, structures, properties, and applications of hydrogels. Necessary recent and relevant papers should be added to support these statements: Regenerable bacterial killing–releasing ultrathin smart hydrogel surfaces modified with zwitterionic polymer brushes; 3D printing hydrogels for actuators: A review; Bio-Inspired Antibacterial Cellulose Paper-Poly (amidoxime) Composite Hydrogel for High-Efficient Uranium (â…¥) Capture from Seawater; etc.

10. Authors cited too few references to support the reasoning, more works should be cited for comparison.

11. There are still some minor typos and grammar issues, such as some spaces that shouldn't be there.

Author Response

Dear reviewer

Thank you for taking the time to review our manuscript. We appreciate your constructive feedback and have revised our work according to your comments. Your insights are valuable and help us improve the quality of our research. In the following document, we will reply point-by-point (in red) to your comments (in italics).

In this manuscript, authors fabricated biocompatible, resilient, and tough nanocellulose tunable hydrogels by introducing mixtures of differing ratios of CNC and CNF. The manuscript is also well-organized. However, there are still some issues to be addressed. A moderate revision is suggested before its acceptance.

  1. The background of this work should be shortened while more solid data should be added into the abstract section.
  • As suggested, the introduction has been shortened to roughly 4 paragraphs. Relevant literature has been referenced to better outline the research context and purpose while sections discussing more general topics such as general information about cellulose and grafting have been discarded.
  1. More introduction on the structures, properties, and applications of nanocellulose and its related hydrogels should be further clarified with some recent supporting articles: Journal of Bioresources and Bioproducts, Volume 7, Issue 4, November 2022, Pages 245-269; Hydrochar-embedded carboxymethyl cellulose-g-poly(acrylic acid) hydrogel as stable soil water retention and nutrient release agent for plant growth; etc.
  • As suggested, the introduction has been rearranged and recent relevant literature has been added to both emphasize the role grafted cellulose composites generally (ref 19-24) and cerium-initiated grafted cellulose composites specifically (ref 28-31).
  1. One scheme to present the experimental procedure should be added for better understanding.
  • As suggested, figure 2 (previously figure 1) was altered to include a schematic description of the hydrogel’s fabrication. Matching explenation added to figure description.
  1. Some of the figures should be modified with better readability, especially the texts.
  • Addressed- imagery and text have been enlarged for better readability (figures 5, 8 and 10). Please notify if additional figures require work.
  1. In Fig.5, the labels in (d) and (e) do not agree with what is written above, and the research object in (f) is not marked.
  • Addressed- Figure 5 has been modified to improve readability and labels reassigned accordingly. Object (f) has been discarded.
  1. Figures should be places under the contents when the figures were firstly mentioned.
  • Addressed- figure 1 and 2 have been switched and noted respectively in the manuscript.
  1. In table 3, the meaning of τy is not explained in the text.
  • Addressed- a clearer explanation was added to the materials and methods section.
  1. The spindle shaped morphology mentioned in the article is not clearly seen in Fig.10.
  • As suggested, the referred sentence has be rephrased to refer the reader to figure 10, a, to observe the spindle shaped morphology in sample C1F3 (as compared to the round/oval morphology and relatively low cell coverage of the control AM sample) and to figure 10, d to observe the migration of cells into the hydrogel.
  1. Authors have introduced the definition, structures, properties, and applications of hydrogels. Necessary recent and relevant papers should be added to support these statements: Regenerable bacterial killing–releasing ultrathin smart hydrogel surfaces modified with zwitterionic polymer brushes; 3D printing hydrogels for actuators: A review; Bio-Inspired Antibacterial Cellulose Paper-Poly (amidoxime) Composite Hydrogel for High-Efficient Uranium (â…¥) Capture from Seawater; etc.
  • As suggested, Relevant recent literature has been added to these sections to support statements.
  1. Authors cited too few references to support the reasoning, more works should be cited for comparison.
  • As suggested, the introduction and discussion sections have been expanded upon to better reflect the results of the article while some note has been given to address and compare similar reported systems along with referenced recent literature.
  1. There are still some minor typos and grammar issues, such as some spaces that shouldn't be there.
  • As suggested, the manuscript has been reworked to remove typos and grammar issues

Reviewer 3 Report

In this manuscript, the author proposed a cerium-initiated graft polymerization method for the preparation of cellulose-derived nanoparticles in the hydrogel matrix, resulting in a nanocomposite hydrogel. The fabricated cellulose-based hydrogel showed high resilience, tensile strength, and extensibility. Cellulose-derived nanoparticles served as multifunctional crosslinkers as well as reinforcing agents. I suggest the publication of this manuscript in Nanomaterials, a high-performance journal if the author can address the following concerns appropriately.

1. The author should simplify the content in the introduction section. There are too many paragraphs in the introduction, which may make the logic unclear. Generally, there are three or four paragraphs in the introduction section, and the author should organic it in a defined routine. Besides, the content in the abstract section can be within 200 words.

2. The author should check carefully the content in the manuscript to avoid errors. For example, the author mentioned “GFP” in the abstract, but what is the abbreviation for this? The author abbreviated “Cellulose nano Fibers” and “Cellulose nano Crystals” as “CNF” and “CNC”, respectively. What is the point of uppercasing “Cellulose” and “Fibers”? Besides, in the sentence “CNC displays amazing mechanical properties, akin to materials such as carbon fibers and Kevlar, with single fibers displaying Young’s modulus of 150GPa and tensile strength of 7.5GPa.”, there should be a space between the number and unite. In the sentence “In this work cerium-initiated graft polymerization was applied to cellulose-derived nanoparticles, namely CNC and CNF, in a polyacrylamide matrix to fabricate nanocomposite hydrogels which display high resilience, tensile strength, and extensibility.”, there should be a comma between “In this work” and “cerium-initiated graft polymerization”. In the sentence “Once propagated, the long chains are further crosslinked by adding N,N’-methylene bisacrylamide (BAM) to form a uniform chemical network of long flexible chains interconnected by homogeneously dispersed cellulose-derived nanoparticles, which act both as a multifunctional crosslinkers and as a reinforcing agent- resulting in a highly extendable, resilient soft polymer hydrogels.”, what is the point of “a multifunctional crosslinkers”? These descriptions made the reviewer very confused.

3. The author utilized cerium, CNC, and CNF for the preparation of hydrogels. Likewise, recently published papers (DOI: 10.1016/j.apsusc.2022.153803; 10.1002/adfm.202107437; 10.1021/acsnano.0c07998; 10.1021/acsami.2c14907; 10.1063/5.0083278; 10.1002/admi.202201137) also utilized conductive polymers, MXene, graphene oxide for hydrogel synthesis. The author can compare the advantages and disadvantages of this hydrogel and those referred ones.

4. The author should ad scale bars to pictures in figures 1 and 2.

Author Response

Dear reviewer

Thank you for taking the time to review our manuscript. We appreciate your constructive feedback and have revised our work according to your comments. Your insights are valuable and help us improve the quality of our research. In the following document, we will reply point-by-point (in red) to your comments (in italics).

In this manuscript, the author proposed a cerium-initiated graft polymerization method for the preparation of cellulose-derived nanoparticles in the hydrogel matrix, resulting in a nanocomposite hydrogel. The fabricated cellulose-based hydrogel showed high resilience, tensile strength, and extensibility. Cellulose-derived nanoparticles served as multifunctional crosslinkers as well as reinforcing agents. I suggest the publication of this manuscript in Nanomaterials, a high-performance journal if the author can address the following concerns appropriately.

  1. The author should simplify the content in the introduction section. There are too many paragraphs in the introduction, which may make the logic unclear. Generally, there are three or four paragraphs in the introduction section, and the author should organic it in a defined routine. Besides, the content in the abstract section can be within 200 words.
  • The introduction has been shortened to roughly 4 paragraphs. Relevant literature has been referenced to better outline the research context and purpose, while sections discussing more general topics such as general information about cellulose and grafting have been discarded.
  • Likewise, the abstract has been shortened to fit the 200 word limitation and rearranged to better summarize the main results presented in the manuscript.
  1. The author should check carefully the content in the manuscript to avoid errors. For example, the author mentioned “GFP” in the abstract, but what is the abbreviation for this? The author abbreviated “Cellulose nano Fibers” and “Cellulose nano Crystals” as “CNF” and “CNC”, respectively. What is the point of uppercasing “Cellulose” and “Fibers”? Besides, in the sentence “CNC displays amazing mechanical properties, akin to materials such as carbon fibers and Kevlar, with single fibers displaying Young’s modulus of 150GPa and tensile strength of 7.5GPa.”, there should be a space between the number and unite. In the sentence “In this work cerium-initiated graft polymerization was applied to cellulose-derived nanoparticles, namely CNC and fibrils, in a polyacrylamide matrix to fabricate nanocomposite hydrogels which display high resilience, tensile strength, and extensibility.”, there should be a comma between “In this work” and “cerium-initiated graft polymerization”. In the sentence “Once propagated, the long chains are further crosslinked by adding N,N’-methylene bisacrylamide (BAM) to form a uniform chemical network of long flexible chains interconnected by homogeneously dispersed cellulose-derived nanoparticles, which act both as a multifunctional crosslinkers and as a reinforcing agent- resulting in a highly extendable, resilient soft polymer hydrogels.”, what is the point of “a multifunctional crosslinkers”? These descriptions made the reviewer very confused.
  • As suggested, the manuscript has been reworked to remove typos and grammar issues. We have reviewed each of the points raised and highly apricate the specific examples given. Appropriate abbreviations have been given where lacking, uppercased letters have been re-examined, appropriate spaces, periods and commas have been added and confusing sentences altered or discarded to better convey the manuscript.  
  1. The author utilized cerium, CNC, and CNF for the preparation of hydrogels. Likewise, recently published papers (DOI: 10.1016/j.apsusc.2022.153803; 10.1002/adfm.202107437; 10.1021/acsnano.0c07998; 10.1021/acsami.2c14907; 10.1063/5.0083278; 10.1002/admi.202201137) also utilized conductive polymers, MXene, graphene oxide for hydrogel synthesis. The author can compare the advantages and disadvantages of this hydrogel and those referred ones.
  • The introduction and discussion sections have been expanded upon to better reflect the results of the article while some note has been given to address and compare similar reported systems along with referenced recent literature.
  1. The author should ad scale bars to pictures in figures 1 and 2.
  • Addressed- scale bars have been added to figures 1 and 3 accordingly.

Round 2

Reviewer 1 Report

Please write "nano fibrils" and "nano crystals" together, i.e. "nanofibrils" and "nanocrystal". The manuscript can be then accepted.

Author Response

Dear Reviewer,
Thank you for taking the time to review our manuscript a second time. Once agin, we appreciate your constructive feedback and have revised our work according to your comments. In the following document, we will reply point-
by-point (in red) to your comments (in italics).

Please write "nano fibrils" and "nano crystals" together, i.e. "nanofibrils" and "nanocrystal". The manuscript can be then accepted.

  • As suggested, the terms "nano fibrils" and "nano crystals" have been grouped together to "nanofibrils" and "nanocrystal" respectively.

Reviewer 2 Report

Authors have revised the manuscript mostly. However, there are still some minor issues to be addressed. A minor revision is still required.

1. More details on the used cellulose nano crystals should be provided. It is better to provide the SEM or TEM images to present the morphology and size.

2. Fig.10 still needs modifications to have a better readability.

3. More highly related papers should be included to support the statements in introduction and discussion: Journal of Bioresources and Bioproducts, Volume 7, Issue 4, November 2022, Pages 245-269; Regenerable bacterial killing–releasing ultrathin smart hydrogel surfaces modified with zwitterionic polymer brushes; Bio-Inspired Antibacterial Cellulose Paper-Poly (amidoxime) Composite Hydrogel for High-Efficient Uranium (â…¥) Capture from Seawater; etc.

Author Response

Dear Reviewer,
Thank you for taking the time to review our manuscript a second time. Once agin, we appreciate your constructive feedback and have revised our work according to your comments. In the following document, we will reply point-
by-point (in red) to your comments (in italics).

Authors have revised the manuscript mostly. However, there are still some minor issues to be addressed. A minor revision is still required.

  1. More details on the used cellulose nano crystals should be provided. It is better to provide the SEM or TEM images to present the morphology and size.
  • As suggested, TEM micrographs of CNC and CNF were added to figure 2,a as part of the hydrogel fabrication overview to provide a better understanding of the materials used.
  1. Fig.10 still needs modifications to have a better readability.
  • As suggested, some text in figure 10,b has been reworked to increase readability.
  1. More highly related papers should be included to support the statements in introduction and discussion: Journal of Bioresources and Bioproducts, Volume 7, Issue 4, November 2022, Pages 245-269; Regenerable bacterial killing–releasing ultrathin smart hydrogel surfaces modified with zwitterionic polymer brushes; Bio-Inspired Antibacterial Cellulose Paper-Poly (amidoxime) Composite Hydrogel for High-Efficient Uranium (â…¥) Capture from Seawater; etc.
  • As suggested, more recent and relevant literature was added and referenced in the introduction and discussion sections.